



**Distribution and stable carbon isotopic composition of dicarboxylic**
**acids, ketocarboxylic acids and α-dicarbonyls in fresh and aged**
**biomass burning aerosols**
Minxia Shen[1,2], Kin Fai Ho[3,4], Wenting Dai[1], Suixin Liu[1], Ting Zhang[1], Qiyuan Wang[1],
Jingjing Meng[5], Judith C. Chow[1,6], John G. Watson[1,6], Junji Cao[1*], Jianjun Li[1,7*]
[1]State Key Laboratory of Loess and Quaternary Geology, Key Lab of Aerosol
Chemistry and Physics, Institute of Earth Environment, Chinese Academy of
Sciences, Xi'an 710061, China;
[2]University of Chinese Academy of Sciences, Beijing, China;
[3]The Jockey Club School of Public Health and Primary Care, The Chinese University
of Hong Kong, Hong Kong, China;
[4]Shenzhen Municipal Key Laboratory for Health Risk Analysis, Shenzhen Research
Institute, The Chinese University of Hong Kong, Shenzhen, China;
[5] School of Geography and the Environment, Liaocheng University, Liaocheng
252000, China;
[6] Division of Atmospheric Sciences, Desert of Research Institute, Reno, Nevada, USA
[7] CAS Center for Excellence in Quaternary Science and Global Change, Xi'an 710061,
China.

*Corresponding author: Jianjun Li, e-mail address:lijj@ieecas.cn
Junji Cao, e-mail address:cao@loess.llqg.ac.cn;





## Abstract

Biomass burning (BB) is a significant source for dicarboxylic acids (diacids) and related compounds that play important roles in atmospheric chemistry and climate change. In this study, a combustion chamber and oxidation flow reactor were used to generate fresh and aged aerosols from burned rice, maize, and wheat straw to investigate atmospheric aging and the stable carbon isotopic ($\delta^{13}C$) composition of these emissions. Succinic acid ($C_4$) was the most abundant species in fresh samples; while, oxalic acid ($C_2$) became dominant after atmospheric aging. Of all diacids, $C_2$ had the highest aged to fresh emission ratios of 50.8 to 64.5, suggesting that $C_2$ is largely produced through secondary photochemical processes. Compared with fresh samples, the emission factors of ketocarboxylic acids and α-dicarbonyls increased after 2-day but decreased after 7-day aging, indicating short residence time and further atmospheric degradation from 2- to 7-days. The $C_2$ $\delta^{13}C$ values for aged biomass samples were higher than those of urban aerosols but lower than marine or mountain aerosols, and the $C_2$ $\delta^{13}C$ became isotopically heavier during aging. Relationships between the reduction in volatile organic compounds (VOCs), such as toluene, benzene, and isoprene, and increase in dicarboxylic acids after 2-day aging indicate that these volatile organic compounds led to the formation of dicarboxylic acids.

*Keywords:* Biomass burning, Dicarboxylic acids, Atmospheric aging, Stable carbon isotope, VOCs



## 1. Introduction


Dicarboxylic acids (diacids), ketocarboxylic acids and α-dicarbonyls are
common components of the atmospheric organic aerosol, accounting for 1–3% of the
total organic carbon in urban areas and >10% of the carbon mass in remote regions
(Kawamura and Usukura, 1993; Kawamura and Sakaguchi, 1999; Kerminen et al.,
2000; Zhao et al., 2018). Due to their high water-solubility and other physicochemical
properties, diacids affect the hygroscopic growth of particulate matter (PM), and these
compounds are involved in the activation of cloud condensation nuclei and formation
of ice nuclei (Kawamura and Bikkina, 2016). Diacids and related compounds have
been found in a wide variety of environments including urban settings (Ho et al., 2006;
Kawamura and Ikushima, 1993; Meng et al., 2020; Sorathia et al., 2018; Wang et al.,
2002, 2006, 2012), mountains ranges (Kawamura et al., 2013; Kunwar et al., 2019),
and remote marine atmospheres (Hoque et al., 2020; Kawamura and Usukura, 1993).
They also have been reported in both the Arctic and Antarctic aerosols (Kawamura et
al., 1996a, b; Narukawa et al., 2002, 2003) as well as polar ice cores (Legrand and De
Angelis, 1996; Kawamura et al., 2001). Various studies have assessed the molecular
distributions, temporal variability, and sources of diacids in different air-sheds.
There are both primary and secondary sources for dicarboxylic acids (Mkoma
and Kawamura, 2013). Primary sources include emissions from fossil fuel combustion
(Kawamura and Kaplan, 1987;Rogge et al., 1993), cigarette burning (Rogge et al.,
1994), cooking (Rogge et al., 1991), and biomass burning (BB) (Narukawa et al.,
1999; Schauer et al., 2001). Of these, BB was found to be an important source of
dicarboxylic acids and related compounds over regional and global scales (Kundu et
al., 2010). Emissions from BB not only compose a major source of primary particles
but also introduce aerosol precursors to the atmosphere (Akagi et al., 2011; Gilman et
al., 2015; Reid et al., 2005). Secondary sources include particles produced by
chemical/photochemical oxidation reactions of volatile organic compounds, especially
those emitted from primary sources (Lim et al., 2013; Carlton et al., 2006, 2007).
Being one of the major contributors to the global budget of aerosols, BB



emissions are of particular concern because they impact air quality, visibility, climate,
and human health (Hodshire et al., 2019). As the largest developing country and one
that burns large quantities of biomass, China has long suffered from severe air
pollution from BB (Chen et al., 2016; Fullerton et al., 2008). Domestic crop residues
(eg. rice, maize, and wheat straw) and firewood are the most significant energy
sources in most rural areas, and these are commonly used for cooking and heating (Li
et al., 2021; Tao et al., 2018).
Although high concentrations of diacids have been detected in biomass
combustion (FaLkovich et al., 2005; Kundu et al., 2010), it is still unclear on the
distribution of diacids directly emitted by BB (Jaffrezo et al., 1998) or compared to
those formed secondarily from percursors (Allen et al., 2004). In addition, limited
data are available on the specific dicarboxylic acids emitted from burning of
agricultural residues. Therefore, it is important to investigate the molecular
composition of dicarboxylic acids in both fresh and aged BB aerosols to advance
current understanding of the potential environmental and climatic effects.
In this study, rice, maize and wheat straw were selected for laboratory
simulations of fresh and aged BB aerosols. The study was conducted with the use of a
combustion chamber and oxidation flow reactor (OFR). Fresh and aged BB aerosols
were chemically analyzed for molecular characteristics and the stable carbon isotopic
composition ($\delta^{13}$C) of selected dicarboxylic acids, ketocarboxylic acids, α-dicarbonyls,
and benzoic. The objectives of this study were to (1) investigate the emissions of
dicarboxylic acids, ketocarboxylic acids and α-dicarbonyls from crop residue burning;
(2) evaluate the effects of atmospheric aging processes on dicarboxylic acids and
related compounds; and (3) investigate reactions of volatile organic carbon
compounds (VOCs) with oxalic acid and intermediates that form in the aging process
to explore potential formation mechanisms of selected organic acids.
**2.   Methods**
**2.1. Preparation and collection of fresh and aged BB aerosols**
Fresh smoke was generated by burning dry biomass fuels (i.e., rice, maize, and



wheat straw) in an ~8 m$^3$ combustion chamber (Tian et al., 2015), and the smoke was
then passed through a Potential Aerosol Mass-Oxidation Flow Reaction (PAM-OFR)
(Aerodyne Research, LLC, Billerica, MA, USA) to simulate aging processes (Cao et
al., 2020). The experimental setup is illustrated in supplementary Fig. S1. Detailed
procedures for sample preparation and collection may be found in previous studies (Li
et al., 2020, 2021; Niu et al., 2020). The PAM-OFR can be used to simulate an
environment with extremely high oxidant concentrations with short residence times
(Kang et al., 2007).
For each test, ~100 g of the biomass fuel was burned on a combustion platform
inside the combustion chamber. Each sampling period lasted 120–180 min, during
which an equal amount of fuel was added to the platform 10 times at regular intervals.
The entire burning cycle, including ignition, flaming, smoldering, and extinction,
intends to simulate real-world source characterization without the use of combustor or
heat preservation. Smoldering was the major driver of the combustion process.
A portion of the diluted smoke was drawn through a quartz fiber filter (47 mm
diameter, Whatman QM/A, Maidstone, UK) at 5 L min$^{-1}$ using a mini-Vol PM$_{2.5}$
sampler (Airmetrics, OR, USA) to capture fresh emission, and another portion (~9 L
min$^{-1}$) was drawn into a 19-L cylinder PAM-OFR (with a diameter of 20 cm and
length of 60 cm) to simulate atmospheric aging. The aging times are selected to
represent lifetimes of regional air pollutants prior to arrival at a receptor (Chow et al.,
2019). Three oxidants (O$_3$, •OH, and •HO$_2$) were generated in the PAM chamber
using irradiation from ultraviolet (UV) lamps. The UV lamps operated at a voltage of
2 and 3.5 V, and the OH exposure values (OH$_{exp}$) in the chamber were estimated at
$2.6 \times 10^{11}$ and $8.8 \times 10^{11}$ molecules-sec/m$^3$, respectively. These levels corresponded
to ~2 and 7 day of aging (Watson et al., 2019), assuming that a representative
atmospheric •OH level of $1.5 \times 10^6$ molecules/m$^3$ (Mao et al., 2009). The aged
aerosols were sampled following the reactions in the PAM-OFR chamber. Each test
was conducted in triplicate to account for experimental errors and to provide a
measure of variability, which was calculated as standard deviations. A total of 36
samples were collected and analyzed for chemical composition.





## 2.2. Sample extraction, derivatization, and quantification

For dicarboxylic acids, ketocarboxylic acids and α-dicarbonyls analyzing, one quarter of each filter sample was extracted three times (15 min each) with purified (18.2 MΩ) water (Milli-Q, Merch, France) and ultrasonication. The pH of the aerosol extracts was adjusted to 8.5 to 9.0 using a 0.1 M potassium hydroxide solution prior to drying that convert carboxylic acids into their salts (Bikkina et al., 2021). This drying step improves the recovery of smaller diacids, such as oxalic acid (Hegde and Kawamura, 2012). Water extracts were concentrated to near dryness with a rotary evaporator under vacuum and then reacted with 14% $BF_3$/n-butanol at 100 ℃ for 1 h to derivatize carboxyl groups to dibutyl esters and oxo groups to dibutoxyacetals.

After derivatization, n-hexane was added and washed with pure water three times to remove the water-soluble inorganics such as hydrogen fluoride and boric acid. The hexane layer was concentrated to near dryness using a rotary evaporator under vacuum and a $N_2$ blow-down technique, and then the esters and acetals of target analytes were dissolved in known amounts of n-hexane. Finally, the hexane layers were concentrated to 100 μL and analyzed using a capillary gas chromatography (GC; HP 6890, Agilent Technology, Santa Clara, CA, USA) equipped with a split/splitless injector and a flame ionization detector (FID). Peak identification was performed by comparing the GC retention times with those of authentic standards and confirmed by a thermal desorption (TD) unit coupled with a gas chromatograph/mass spectrometric detector (TD-GC/MS, Models 7890A/5975C, Agilent Technology, Santa Clara, CA, USA). The detection limits for those organic compounds were 0.1 ng m$^{-3}$, and the analytical errors, based on the replicate analyses, were less than 15%. Recoveries of the target compounds were 83% for oxalic acid and 87% to 110% for the other species.

## 2.3 Emission factor calculations

Concentrations of the various species in the aged samples were affected by their initial emission, also undergo degradation and production through secondary chemical processes. Fresh and aged fuel-based emission factors (EF) for each measured chemical compound were calculated by dividing its filter mass by the mass of





combusted dry biomass fuel (Andreae and Merlet, 2001; Li et al., 2020; Tian et al.,
2015); that is:

$$EF_i = \frac{m_i \times v_{Stk} \times D \times t_{sample}}{Q_p \times m_{fuel}} \times DR$$

where $EF_i$ (mg kg$^{-1}$) is the EF of chemical compound i for the specific crop; $m_i$ (mg)
is the mass of chemical compound i collected on the filter; $v_{Stk}$ is the average stack
flow velocity (m s$^{-1}$) at standard conditions; D is the stack cross section (m$^2$); $t_{sample}$ is
the sampling duration (s); $Q_p$ is the sampling volume through the filter (m$^3$) at
standard temperature and pressure; and $m_{fuel}$ is the mass of burned biomass fuel (kg,
dry weight).
The dilution ratio (DR) was determined from the $CO_2$ concentrations measured at
the stack, diluted stack, and background, where:

$$DR = \frac{CO_{2,Stk} - CO_{2,Bkg}}{CO_{2,Dil} - CO_{2,Bkg}}$$

where $CO_{2,Stk}$ is the $CO_2$ concentration in the stack; $CO_{2,Bkg}$ the background $CO_2$
concentration in the atmosphere; and $CO_{2,Dil}$ the $CO_2$ concentration in the diluted
smoke.

**2.4. Stable carbon isotope composition of dicarboxylic acids**

Stable carbon isotopic determinations ($\delta^{13}$C) of dicarboxylic acids,
ketocarboxylic acids, and α-dicarbonyls followed the techniques of Kawamura and
Watanabe (2004). The isotope values of the derivatized samples were determined
using a gas chromatography–isotope ratio mass spectrometer (GCIR-MS; Thermo
Fisher, Delta V Advantage, Franklin, MA, USA). The $\delta^{13}$C values were then
calculated for free organic acids using an isotope mass balance equation based on the
measured $\delta^{13}$C values of derivatives and the derivatizing agent (BF$_3$/n-butanol)
(Kawamura and Watanabe, 2004). To ensure the analytical error of the $\delta^{13}$C values
less than 0.2‰, each aerosol sample was analyzed in triplicate, to obtain the average
values.



## 3. Results and Discussion

### 3.1. Emission factors for dicarboxylic acids, ketocarboxylic acids, α-dicarbonyls

Fresh and aged $PM_{2.5}$ EFs for a homologous series of dicarboxylic acids, ketocarboxylic acids (glyoxylic acid, $\omega C_2$ and pyruvic acid, Pyr), α-dicarbonyls (glyoxal, Gly and methylglyoxal, mGly) and benzoic acid are presented in Table 1. The EFs for most fresh and aged diacids varied by severely order-of-magnitude with higher EFs after atmospheric aging. The highest fresh EF (i.e. $EF_{fresh}$) was found for wheat straw ranging 44 - 122 mg kg$^{-1}$ for succinic acid and 67-102 mg kg$^{-1}$ for glyoxal, higher than those found in maize and rice. The arithmetic means and standard deviations for the $EF_{fresh}$ of total dicarboxylic acids from burning of rice, maize, and wheat straws were 84 ±36, 130 ±47, and 307 ±141 mg kg$^{-1}$, respectively.

As is shown Fig. 1, distributions of dicarboxylic acids in fresh emissions varied by crop types and species. Of the saturated n-dicarboxylic acids, succinic acid ($C_4$) acid was the most abundant species in the maize and wheat straw with average $EF_{fresh}$ of 22 ±12 and 83 ±46 mg kg$^{-1}$, respectively. Azelaic acid ($C_9$) and $C_4$ were the most abundant species from rice burning with $EF_{fresh}$ of 11 ± 2.9 and 10 ± 9.0 mg kg$^{-1}$, respectively. These findings are consistent with the fresh smoke aerosols in Siberian BB plumes (Kalogridis et al., 2018), in which $C_4$ and $C_9$ were more abundant than oxalic acid ($C_2$). Previous studies also showed $C_9$ to be an oxidation product of unsaturated fatty acids in biomass smoke (Kawamura and Gagosian, 1987; Kawamura et al., 2013; Agarwal et al., 2010; Cao et al., 2017).

Similar to the diacids, the highest $EF_{fresh}$ for ketocarboxylic acids and α-dicarbonyls were also found in wheat straw samples, with 44 ±31 and 138 ±91 mg kg$^{-1}$, respectively. Glyoxal (Gly) was the highest α-dicarbonyls, with average $EF_{fresh}$ of 27 ± 3.9, 42 ± 10, and 84 ± 41 mg kg$^{-1}$ for rice, maize and wheat straw, respectively. This is consistent with previous studies which showed that Gly is often more abundant than methylglyoxal (mGly) in polluted aerosols collected from China (Pavuluri et al., 2010; Ho et al., 2007). Benzoic acid also was determined, and its $EF_{fresh}$ for rice, maize, and wheat aerosols were 1.9 ±0.2, 2.5 ±0.4, and 3.1 ±0.3 mg





kg$^{-1}$ (Table 1).

### 3.2. Effects of atmospheric aging processes

3.2.1 Dicarboxylic acids
The EF$_{aged}$ of 2- and 7-day diacids were 1650 $\pm$ 438 and 1957 $\pm$ 598 mg/kg,
respectively (Table S1); approximately 10 times greater than the EF$_{fresh}$. High
aged/fresh ratios implies that the bulk of the total dicarboxylic acids were secondarily
produced through aging processes. Longer exposure time in the atmosphere increased
the formation of diacids as ratios of average aged/fresh increased from 9.1 (2-day) to
10.8 (7-day) (Table S1).
As shown in Fig. 2, oxalic acid (C$_2$) was the most abundant of all measured
diacids among three crops, with the highest EF$_{aged}$ found in wheat (1412 $\pm$ 328 mg/kg)
after 7-day aging. The aged/fresh (A/F) ratios for oxalic acid increased by ~27% from
50.8 (2-day) to 64.5 (7-day) (Table S1). These results are further evidence that PM$_{2.5}$
oxalic acid is largely produced by secondary photochemical processes rather than
direct emissions in biomass burning. This also is a likely reason why C$_2$ is often the
most abundant diacid in ambient samples, especially in the oceanic and other remote
areas (Hoque et al., 2020;Kawamura and Usukura, 1993;Kawamura and Sakaguchi,
1999;Kunwar and Kawamura, 2014;Hegde and Kawamura, 2012;Kawamura and
Bikkina, 2016;Wang et al., 2012). Two-day aging appeared to be sufficient for maize,
with degradation after 7-day. This may partial due to the low VOCs produced in
maize burn (Niu et al., 2020). The A/F ratios in Fig. 3 showed that 2-day aging is
sufficient for many of the diacids.
Succinic acid (C$_4$) ranked second in abundance after C$_2$, with 7-8 folds increased
in EF after 2- and 7-day aging wheat. Although malonic acid (C$_3$) is mainly produced
by the photochemical oxidation of succinic acid, it also can be formed through the
incomplete combustion of fossil fuels and biomass (Kawamura and Ikushima, 1993).
In the atmosphere, C$_4$ is typically more abundant than C$_3$ originated from BB,
vehicular engine exhaust and biogenic emissions (Fu et al., 2013; Kawamura and
Kaplan, 1987; Kundu et al., 2010). Fig. 3 shows atmospheric aging increased the
abundances of C$_3$ and C$_4$ with A/F ratios increased from 16.2 to 31.1 for malonic acid,



and from 5.7 to 8.0 in succinic acid from 2- to 7-day of aging (Table S1). These
findings add to the evidence that these diacids are produced by the photo-oxidation of
primary pollutants emitted from combustion process. Lower EFs and higher A/F ratios
in aged and fresh malonic acid than those of succinic acid may be attributed to rapid
oxidation rate of $C_3$ or decarboxylation processing of $C_4$ diacid during aging (Zhao et
al., 2018).

254         As mentioned above, $C_9$ (azelic acid) is thought to be mainly formed through the

photochemical oxidation of unsaturated fatty acids emitted by plants (Kawamura and
Gagosian, 1987). Average EFs in azelic acid were low, ranging from 18 ±7.3 mg kg$^{-1}$
(fresh), to 51 ± 14 mg kg$^{-1}$ (2-day), with A/F ratios of $C_9$ of 2.8, and 2.2 for the 2- and
7-day samples, respectively, suggesting that azelic acid is relatively stable with short
residence time. Fig.3 shows that A/F ratios of other long-chain dicarboxylic acids and
branched dicarboxylic acids did not show apparent changes between the 2- and 7-day
samples, which may be due to the degradation of long-chain dicarboxylic acids
(Enami et al., 2015;Legrand et al., 2007;Miyazaki et al., 2010). It is also possible that
the laboratory combustion experiment did not produce adequate quantities of certain
diacids. For example, $C_5$ and $C_6$ are commonly formed by reactions of cycloolefins
emitted from anthropogenic sources with $O_3$ (Hatakeyama et al., 1985), and phthalic
acid as a product of the photochemical oxidation of aromatic hydrocarbon compounds
(Kawamura and Ikushima, 1993). Additional laboratory experiments may be needed
to reify different atmospheric process.
3.2.2 Ketocarboxylic acids and α-carbonyls

270         In contrast to the dicarboxylic acids, aging process were not apparent in

ketocarboxylic acids as A/F ratios reduced by 16% from 13.8 (2-day) to 11.9 (7-day).
Similar phenomenon was found for α-carbonyls with A/F ratios reduced by 64% from
5.4 (2-day) to 3.3 (7-day). This suggests the possibility that the degradation of these
intermediates to oxalic acid is faster than their formation by oxidation after 2 days of
aging. Fig.3 also show apparent reduction EF of 33-42% from 2- to 7-day aging for
glyoxal (Gly) and methylglyoxal (mGly) which may be due to the fact that both Gly
and mGly initially can be oxidized to less volatile polar organic acids including



pyruvic acid (Pyr) and glyoxylic acid ($\omega C_2$) and then further oxidized to $C_2$ (Wang et
al., 2012;Warneck, 2003).

**3.3. Comparisons of diagnostic ratios of dicarboxylic acids in fresh and aged**

**aerosols**

Patterns in the relative abundances of dicarboxylic acids have been used to
evaluate biogenic versus anthropogenic source strengths and the photochemical
processing of organic aerosols (Kawamura et al., 2012). Previous studies have shown
that $C_4$ can be directly oxidized into $C_2$ or via $C_3$ into $C_2$ (Jung et al., 2010;
Sorooshian et al., 2007), with $C_2$ being an end-product of the photochemical oxidation
(Wang et al., 2012). The ratios of $C_3/C_4$, $C_2/C_4$ and $C_2$/total diacids can be regarded as
indicators of aerosol aging (Cheng et al., 2013; Kunwar et al., 2019; Meng et al., 2018;
Pavuluri et al., 2010), with higher ratios indicative of more aged aerosols (Kawamura
and Sakaguchi, 1999). As shown in Table 2, the ratios in this study showed a clear
atmospheric aging trend from fresh to 7-day aging with ratios of 0.7 to 6.4 for $C_2/C_4$,
0.1 to 0.6 in $C_2$/total diacids and 0.2 to 0.5 in $C_3/C_4$ with photochemically oxidization
rune pronounced.
Ratios of $\omega C_2/C_2$ and Gly/mGly can also be used to evaluate the oxidation of
organic aerosols (Cheng et al., 2013, 2015; Kawamura et al., 2013). In the study,
apparent reduction of the $\omega C_2/C_2$ ratios from 1.3 (fresh) to 0.2 (7-day) supports the
potential oxidation pathways from precursor glyoxylic to oxalic acids. Similarly, the
Gly/mGly ratios in the biomass burning samples were higher in the fresh $PM_{2.5}$
samples (3.8) compared to the 2-day aged (2.3) and 7-day (2.0) aging, indicating the
degradation for Gly proceeds more rapidly than mGly, and that is consistent with the
decline Gly/mGly ratios in aged aerosols (Cheng et al., 2013).
Ratios of $C_3/C_4$, $C_2$/diacids, $\omega C_2/C_2$, and Gly/mGly are similar among studies.
Except for the higher $C_3/C_4$ ratio of 3.9 found in marine aerosols of over the pacific
region (Kawamura and Sakaguchi, 1999), and lower $C_3/C_4$ ratios in Siberian biomass
burning emissions in a large aerosol chamber (<0.03) (Kalogridis et al., 2018). The
largest difference was found for $C_2/C_4$, varied from <1 for fresh aerosol in Siberian
biomass burning (Kalogridis et al., 2018), to 25.2 from forest fire in Thailand


(Boreddy et al., 2020). Elevated $C_2/C_4$ ratios exceeding 10 were found in aged
ambient Xi'an, China (10.4) (Cheng et al., 2013), Mt. Hua, China (10.7) (Meng et al.,
2014), marine aerosol, Pacific ocean (14.3) (Kawamura and Sakaguchi, 1999), and
ambient island Okinawa, Japan (15.5) (Kunwar and Kawamura, 2014). These $C_2/C_4$
ratios are ~63% to 142% higher than these reported in this study. Overall, these
comparisons show the importance of photochemical aging, however, the atmospheric
oxidation evidently was more extensive in aerosols from some remote mountain and
marine environments.
**3.4. Stable carbon isotopes**
Stable carbon isotope ratios ($\delta^{13}C$) can provide insights into the sources of
aerosols, Pavuluri and Kawamura (2016) reported that average $\delta^{13}C$ values for $C_2$
from biogenic aerosols (-15.8‰) were less negative—i.e., contained more $^{13}C$ and
was isotopically enriched than those from anthropogenic aerosols (-19.5‰). Data for
$\delta^{13}C$ also can provide information on the processing or aging of organic aerosols
because isotopic fractionation result from chemical reactions or phase transfer
(Pavuluri and Kawamura, 2016; Zhang et al., 2016). Mass loading of $\delta^{13}C$ for diacids
in the fresh BB samples were too low to be detected by the GCIR-MS, but the $\delta^{13}C$
values for $C_2$ ranged from -23.3 to -21.0 ‰ (with an average of -21.9 ± 1.2 ‰) in
2-day and -19.1 to -15.5 ‰ (-17.3 ± 1.7 ‰) for 7-day aged samples (Table 3).
Table 3 shows that aged maize samples reported the a heaviest $\delta^{13}C$ signatures
than those of rice and wheat. This is likely because maize is a $C_4$ plant, whereas wheat
and rice are both $C_3$ plants. Song et al. (2018) showed that $\delta^{13}C_{TC}$ in $C_4$ plants is
isotopically heavier than in $C_3$ plants. Moreover, the $\delta^{13}C$ of $C_2$ is more abundant in 7-
than 2-day samples (Table 3) with -13.1 ± 1.6 ‰ (2-day) and -7.1 ± 1.4 ‰ (7-day) in
maize; -26.2 ± 1.8 ‰ (2-day) and -20.8 ± 3.3 ‰ (7-day) in rice and -26.5 ± 0.2 ‰
(2-day) and -24.0 ± 0.5 ‰ (7-day) in wheat combustion. The $\delta^{13}C$ data for $C_3$, $C_4$ and
$\omega C_2$ (Table S2) showed similar trends, consistent with previous studies. For example,
Zhao et al. (2018) found that the $\delta^{13}C$ values of $C_2$ were related to aging. Pavuluri and
Kawamura (2016) analyzed diacids, $\omega C_2$, and Gly for $\delta^{13}C$ in anthropogenic and
biogenic aerosol samples by UV irradiation, and reported more $\delta^{13}C$ less negative



with longer irradiation times. During atmospheric oxidation reactions, organic
compounds react with OH radicals, causing the release of $CO_2$ and CO which contain
relatively the lighter $^{12}C$ isotope and thus leaving the remaining substrate enriched in
$^{13}C$ (Hoefs, 1997; Sakugawa and Kaplan, 1995).

342         A comparison of $\delta^{13}C$ values for $C_2$ in the aerosols from selected environments is

shown in Figure 4. Average $\delta^{13}C$ value (-21.9 ± 1.2 ‰) of 2-day aged biomass
burning of $C_2$ was comparable to those reported for urban regions, such as Beijing
(-21.8 ± 2.8‰) (Zhao et al., 2018) and Liaocheng (-19.8 ± 3.1‰) (Meng et al., 2020)
(Table 3). With continued aging, the $C_2$ $\delta^{13}C$ of the 7-day samples (-17.3 ± 1.7 ‰)
was more similar in samples from Mt. Tai (-16.5 ± 1.8‰) (Meng et al., 2018) and
western pacific and southern ocean aerosol (-16.8 ± 0.8‰) (Wang and Kawamura,
2006), but it was significantly lighter than that of samples from the Korea Climate
Observatory at Gosan (-13.7 ± 2.5‰), which is a mountain background site in East
Asia (Zhang et al., 2016).
**3.5. Relationships between volatile organic carbon compounds and dicarboxylic**
**acids**

354         During the chamber experiment (Niu et al., 2020) concerning measured the VOC

compounds. Table S3 presents the correlations between decreases in VOCs and
increases in dicarboxylic acids from fresh to 2-day aged BB samples. Significant
(0.01 <p < 0.05) correlations (R) were observed for toluene with Gly (R = 0.75),
mGly (R = 0.81), Pyr (R =0.78), $\omega C_2$ (R = 0.78) and $C_2$ (R = 0.67) (Fig. 5), suggesting
that toluene was converted to diacids during the aging processes. Indeed, it has been
reported that the photooxidation of toluene is a potential source of secondary organic
aerosol (SOA) in urban air (Sato et al., 2007), and the major chemical components of
the SOA include hemiacetal, peroxy hemiacetal oligomers and dicarboxylic acids. It
also can be seen that benzene had significant correlations with mGly and $C_2$ (R>0.59
in Fig.5), implying that the oxidation of benzene led to diacid formation. And we can
see that the slope of the correlation between the decrease of toluene and benzene and
the increase of the precursor (Pyr and $\omega C_2$) is significantly higher than that of oxalic
acid.





On the global scale isoprene is the most important precursor for $C_2$, contributing
70% to the global $C_2$, while anthropogenic VOCs contribute 21% to $C_2$ production
(Myriokefalitakis et al., 2011). Thus, it is not surprising that isoprene correlated with
$C_2$ (R=0.58) (Fig.5). In addition, several alkenes and alkanes also had a significant
correlation with $C_2$ (Table S3), indicating that these species may react in secondary
oxidation processes to generate oxalic acid. Previous studies have confirmed that
dicarboxylic acids can be oxidation products of aromatic hydrocarbons (Borrás and
Tortajada-Genaro, 2012), cycloolefins (Hamilton et al., 2006), and may originated
from diesel vehicle exhaust (Samy and Zielinska, 2010). However, no significant
correlation was found between decreases in VOCs and increases in 7-day aged
dicarboxylic acids. For the longer aging times, the particulate phase compounds may
be further oxidized to generate other compounds besides diacids.
**4.  Conclusions**
The emission factors (EFs) of dicarboxylic acids and related compounds in
experimentally produced fresh and aged biomass burning aerosols were compared.
For fresh emissions, succinic acid ($C_4$) was the most abundant diacid species followed
by azelaic acid ($C_9$). After atmospheric aging, $C_2$ dominated the diacids, with elevated
emission factors. Ratios of aged to fresh (A/F) for $C_2$ increased from 50.8 (2-day) to
64.5 (7-day). These results suggest that the dicarboxylic acids in the atmosphere
largely originated from secondary photochemical processes as opposed to primary
emissions from biomass burning. The 2-day A/F ratios 2.8 of azelaic acid ($C_9$)
degraded by 27% after 7-days, suggesting that this species is relatively stable with
short residence time.
Decreasing trends in EFs were found for ketocarboxylic acids and α-dicarbonyls,
from 2-day to 7-day aging with A/F ratios reduced from 13.8 to 11.9 and from 5.4 to
3.3, respectively. These results suggest that after 2-day aging, the net degradation of
these intermediates was faster than their rates of formation. Compared with 2-day
samples, the $\delta^{13}C$ of $C_2$, $C_3$, $C_4$ and $\omega C_2$ in 7-day samples became more positive or
isotopically heavier after the additional aging, likely due to kinetic isotope



fractionation effects. Moreover, the $\delta^{13}C$ values for the aged maize samples in both
the 2- and 7-day samples were significantly more positive than those of rice and
wheat. This is likely due to differences in their photosynthetic pathways as maize is a
$C_4$ plant, while wheat and rice are both $C_3$ plants. The correlations between VOCs
(benzene, toluene, isoprene, etc.) and oxalic acid ($C_2$) and intermediates indicated that
the oxidation of VOCs led to the formation of diacids. The results provide in-depth
understanding of SOAs formation in regions greatly affected by biomass burning.

**Data availability**

The data involved in this study will be provided when they are asked from the
corresponding authors.

**Author contribution**

Junji Cao and Jianjun Li conceived and designed the study. Minxia Shen
contributed to the literature search, samples and data analysis, and manuscript writing.
Jianjun Li, Junji Cao, Judith C. Chow and John G. Watson contributed to manuscript
revision. Kin Fai Ho, Wenting Dai, Suixin Liu, Ting Zhang, Qiyuan Wang, Jingjing
Meng carried out the particulate samples and supervised the experiments. All authors
commented on the manuscript and reviewed the manuscript.

**Declaration of Competing Interest**

The authors declare that they have no known competing financial interests or
personal relationships that could have appeared to influence the work reported in this
paper.

**Acknowledgments**

This work was jointly supported by the program from National Nature Science
Foundation of China (No. 41977332), the Strategic Priority Research Program of
Chinese Academy of Sciences (No. XDB40000000), the Innovation Capability
Support Program of Shaanxi (No. 2020KJXX-017), and by the US National Science
Foundation (AGS-1464501 and CHE 1214463). Jianjun Li also acknowledges the
support of the Youth Innovation Promotion Association CAS (No. 2020407).



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

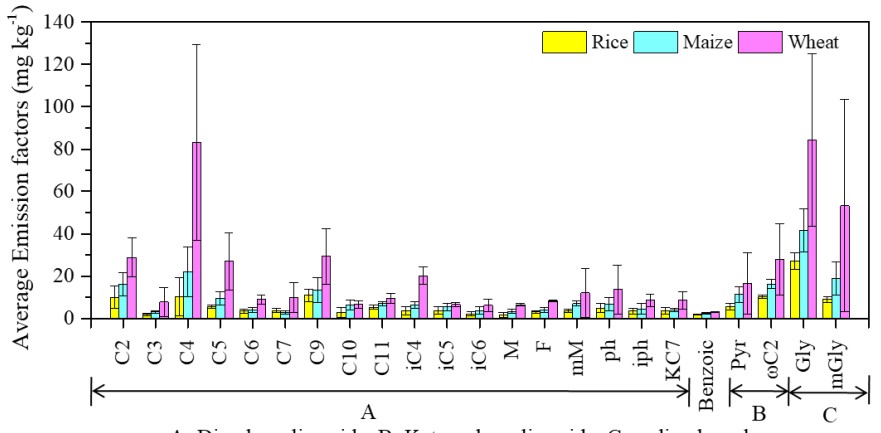

A: Dicarboxylic acids; B: Ketocarboxylic acids; C: α-dicarbonyls


Figure 1 Average emission factors of dicarboxylic acids and related compounds in
719                    fresh PM₂.₅ aerosols from biomass burning.

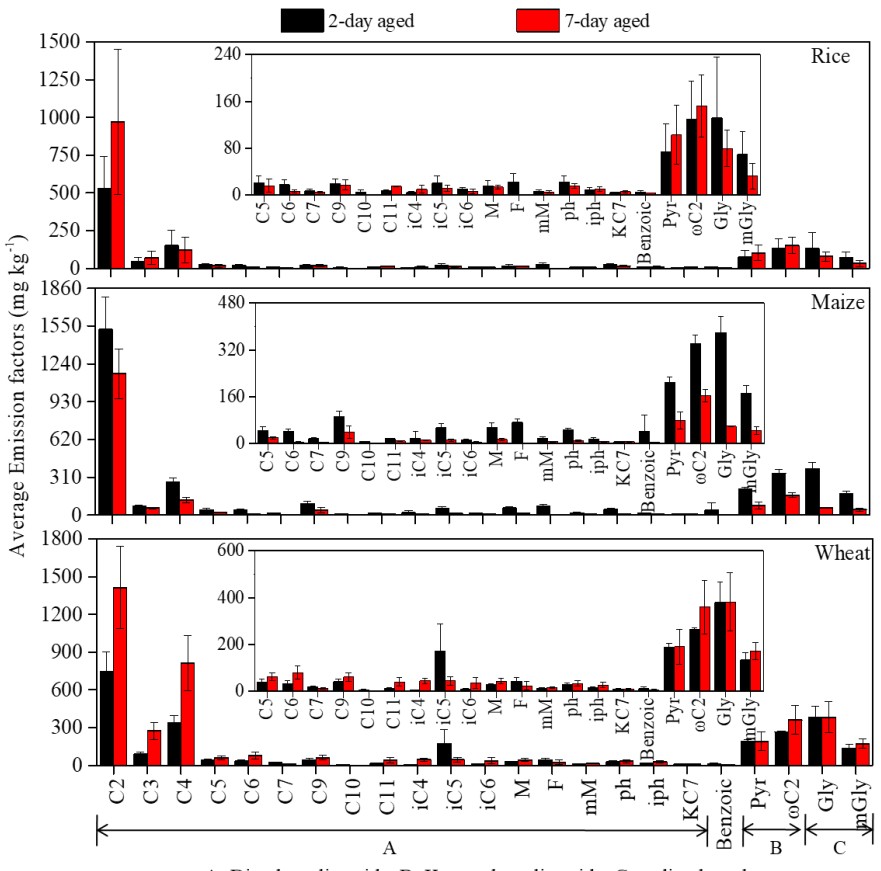

A: Dicarboxylic acids; B: Ketocarboxylic acids; C: α-dicarbonyls

Figure 2 Comparison between 2- and 7-day aged average PM$_{2.5}$ emission factors of A: dicarboxylic acids, B: ketocarboxylic acids, and C: α-carbonyls for laboratory combustion of rice, maize, and wheat straw.

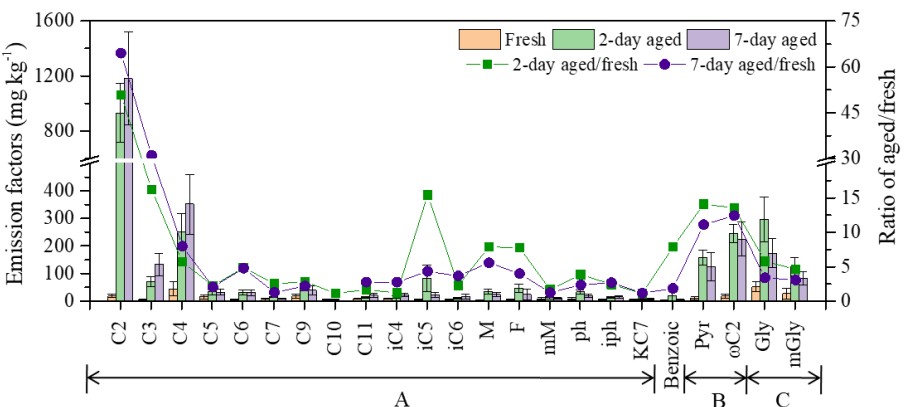

A: Dicarboxylic acids; B: Ketocarboxylic acids; C: α-dicarbonyls

Figure 3 Average emission factors of dicarboxylic acids and related compounds from biomass burning experiment for the fresh, 2- and 7-day aged $PM_{2.5}$ aerosols. The squares and dots denote the ratios of aged to fresh (A/F) sample for the dicarboxylic acids and related compounds after 2- and 7-day aging.





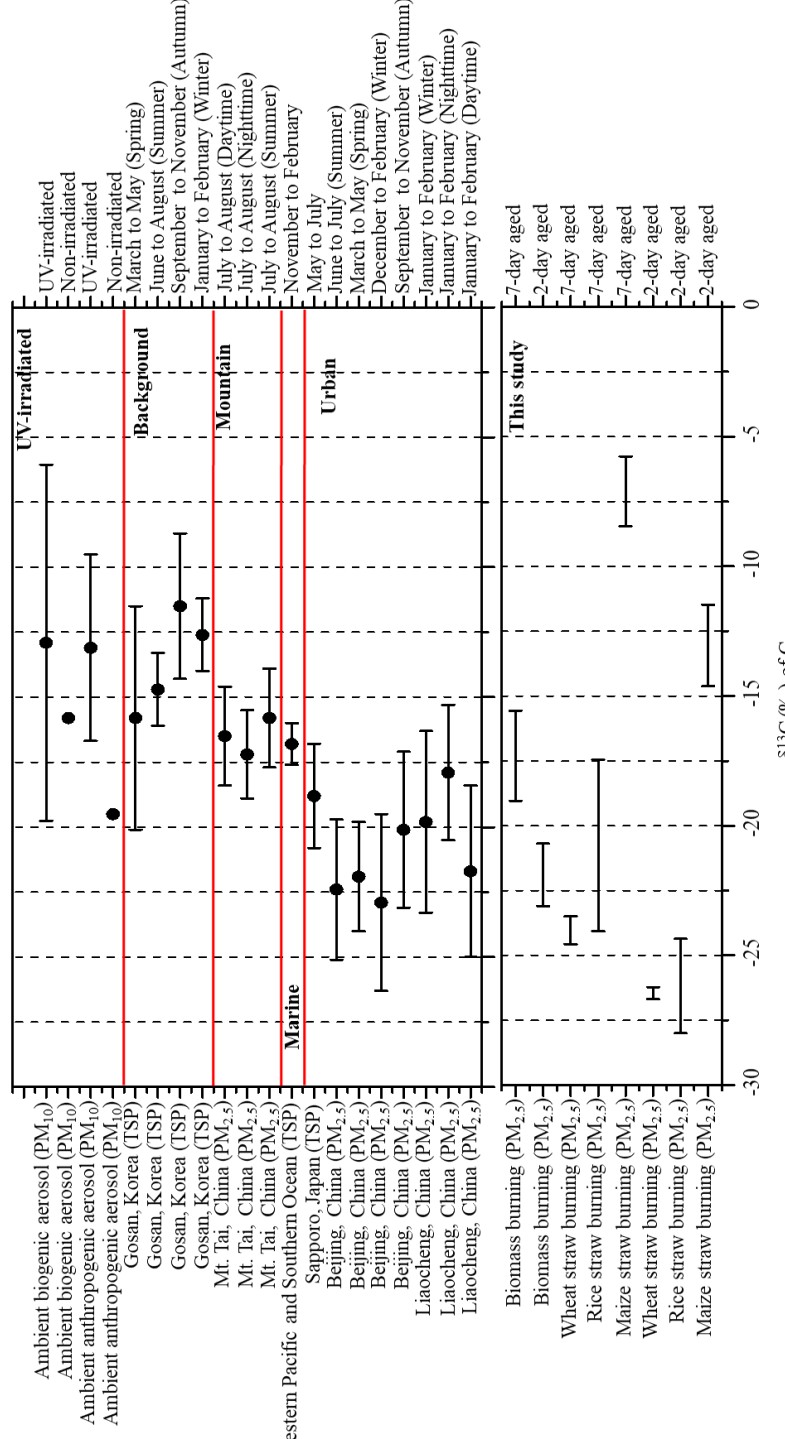

Figure 4 Stable carbon isotope ratios ($\delta^{13}C$, ‰) of $C_2$ in aerosols from selected environments.




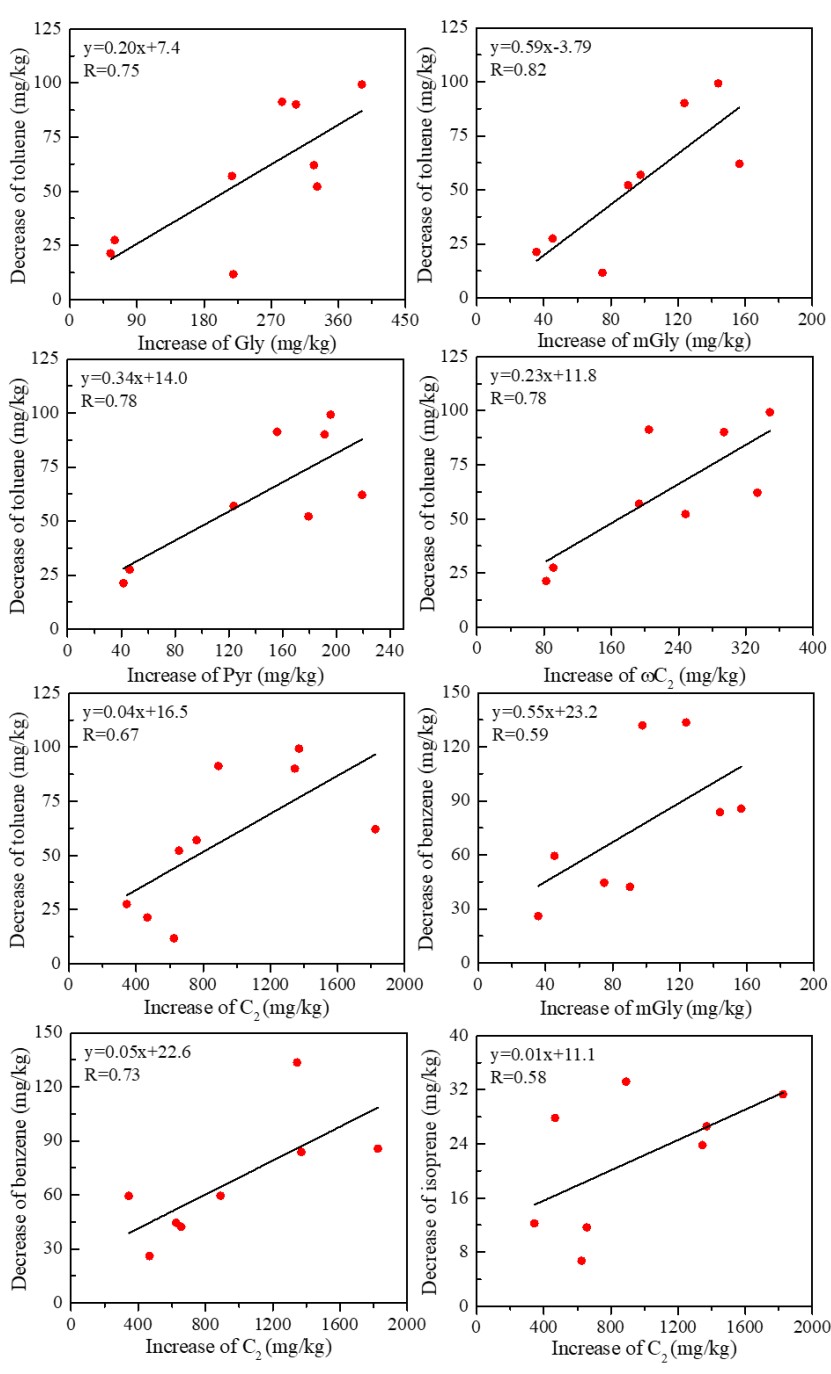


Figure 5 Regressions between the decreases of specific VOCs (toluene, benzene and
isoprene) and increases of $C_2$ and its intermediates methylglyoxal (mGly), Glyoxal,
(Gly), Pyruvic acid (Pyr) and Glyoxylic ($\omega C_2$).



Table 1 Emission factors (EFs, mg kg$^{-1}$) of fresh and aged dicarboxylic acids and related compounds from rice, maize and wheat straw burning.

| Compounds | Rice-2 day aged | | Rice-7 day aged | | Maize-2 day aged | | Maize-7 day aged | | Wheat-2 day aged | | Wheat-7 day aged | |
| --- | --- | --- | --- | --- | --- | --- | --- | --- | --- | --- | --- | --- |
| | Fresh | 2-d aged | Fresh | 7-d aged | Fresh | 2-d aged | Fresh | 7-d aged | Fresh | 2-d aged | Fresh | 7-d aged |
| **I. Dicarboxylic acids** | | | | | | | | | | | | |
| Oxalic, $C_2$ | 5.1±0.9 | 527±214 | 15±10 | 971±482 | 8.1±2.2 | 1522±268 | 24±8.8 | 1158±202 | 18±15 | 742±160 | 39±3.4 | 1412±328 |
| Malonic, $C_3$ | 2.4 | 46±26 | 1.4±0.7 | 70±42 | 3.6 | 74±8.4 | 2.7±0.6 | 56±9.0 | 12±13 | 89±8 | 3.6±0.8 | 273±70 |
| Succinic, $C_4$ | <DL[a] | 152±100 | 10±9.0 | 120±85 | 9.3±12 | 268±35 | 35±12 | 124±23 | 44±72 | 335±62 | 122±21 | 813±217 |
| Glutaric, $C_5$ | <DL | 21±12 | 5.4±0.8 | 16±11 | 8.9 | 44±13 | 10±3.1 | 20±2.8 | 28±23 | 41±12 | 27±3.7 | 61±16 |
| Adipic, $C_6$ | <DL | 18±8.6 | 3.5±0.9 | 6.4±2.2 | <DL | 42±6.4 | 4.1±1.1 | 5.8±0.7 | 12 | 33±11 | 5.5±2.2 | 79±28 |
| Pimelic, $C_7$ | 5.0 | 7.4±2.7 | 2.4±0.9 | 4.9±1.7 | <DL | 18±2.1 | 2.8±0.9 | 5.5±0.3 | 16±12 | 21±3.1 | 4.0±1.6 | 13±2.8 |
| Azelaic, $C_9$ | 11±1.8 | 19±8.8 | 11±3.9 | 18±8.4 | 10±3.2 | 91±21 | 17±8.4 | 39±22 | 23±20 | 41±12 | 35±6.5 | 61±19 |
| Sebacic, $C_{10}$ | 2.8±2.3 | 5.0±4.6 | <DL | <DL | 6.2±2.3 | 7.0±0.5 | <DL | <DL | 6.6±1.7 | 5.7±3.1 | <DL | <DL |
| Undecanedioic, $C_{11}$ | <DL | 7.6±1.5 | 5.4±1.1 | 15±1.2 | 6.7±1.2 | 18±1.5 | 7.5±0.8 | 8.9±2.1 | 11.2 | 14±2.8 | 7.9±2.4 | 40±19 |
| Methylmalonic, $iC_4$ | 3.6 | 4.8±1.8 | 3.7±1.9 | 10±6.8 | 3.8±0.2 | 19±22 | 9.1±2.9 | 11±2.1 | <DL | 5.7±1.4 | 20±4.2 | 46±12 |
| Mehtylsuccinic, $iC_5$ | <DL | 20±13 | 3.8±1.7 | 12±5.5 | <DL | 54±16 | 5.6±1.8 | 12±3.5 | <DL | 172±14 | 6.6±1.0 | 45±19 |
| Methylglutaric, $iC_6$ | <DL | 9.8±2.7 | 2.1±1.0 | 6.1±3.6 | 3.4 | 12±3.4 | 4.0±1.9 | 5.7±1.4 | 7.4±4.9 | 8.3±2.9 | 5.1±1.0 | 37±23 |
| Maleic, M | <DL | 16±8.9 | 1.6±1.2 | 14±3.5 | 2.8±1.1 | 56±14 | 4.0±1.0 | 14±3.6 | 9.6 | 29±3.8 | 3.4±0.6 | 43±11 |
| Fumaric, F | <DL | 22±15 | 3.1±0.6 | <DL | <DL | 73±13 | 4.0±1.1 | <DL | 13 | 43±15 | 3.6±0.3 | 24±19 |
| Methylmaleic, mM | 4.5±0.7 | 6.7±2.1 | 2.5±0.8 | 5.6±2.4 | 7.3±0.5 | 18±5.3 | 6.6±1.9 | 6.5±1.2 | 19±22 | 12±3.4 | 5.5±0.8 | 16±4.7 |
| Phthalic, Ph | 4.0±0.5 | 23±10 | 5.8±3.5 | 16±4.3 | 3.8±1.0 | 47±6.5 | 10±5.4 | 11±2.6 | 10±12 | 29±6.9 | 17±12 | 33±12 |
| Isophthalic, iPh | 4.1 | 8.7±3.8 | 2.9±1.3 | 11±3.6 | 3.9 | 17±3.9 | 5.2±2.7 | 7.3±1.3 | 9.7±2.7 | 16±2.6 | 7.6±3.1 | 27±11 |


| | | | | | | | | | | | | |
|---|---|---|---|---|---|---|---|---|---|---|---|---|
| Ketopimelic, kC$_7$ | <DL | 4.4±0.6 | 3.6±1.6 | 6.0±1.8 | <DL | 6.5±1.9 | 3.9±0.7 | 6.3±0.6 | 13±7.6 | 9.3 ±2.5 | 4.5 ±0.4 | 8.9±3.1 |
| Subtotal | 43±6.2 | 919±437 | 83 ±41 | 1300±665 | 78 ±23 | 2386±440 | 155 ±55 | 1491±279 | 252±206 | 1645±437 | 318±64 | 3032±814 |
| **II. Ketocarboxylic acids** | | | | | | | | | | | | |
| Pyruvic acid, Pyr | 4.6 | 74±48 | 6.5 ±3.1 | 103±50 | 8.1±3.5 | 210±17 | 15±4.3 | 79±29 | 21±25 | 189±15 | 13±4.0 | 190±75 |
| Glyoxylic, ωC$_2$ | 11±0.3 | 129±65 | 9.6±1.3 | 152±53 | 16±2.0 | 341±30 | 17±2.5 | 164±21 | 33±27 | 265±4.9 | 23±6.3 | 359±114 |
| Subtotal | 16±0.4 | 203±113 | 16±4.4 | 255±103 | 24±5.5 | 551±48 | 32±6.8 | 243±50 | 53±52 | 454±20 | 35±10 | 550±189 |
| **III. α-Dicarbonyls** | | | | | | | | | | | | |
| Glyoxal, Gly | 32±1.1 | 132±104 | 22±6.7 | 79±31 | 39±8.6 | 380±54 | 44±12 | 60±1.7 | 102±71 | 380±87 | 67±11 | 382±125 |
| Methylglyoxal, mGly | 15±0.5 | 70±39 | 2.8±2.2 | 33±22 | 30±13 | 172±28 | 7.6 ±2.6 | 46±13 | 91±96 | 135±31 | 16±4.0 | 172±37 |
| Subtotal | 47±1.6 | 202±143 | 25±8.9 | 112±53 | 69±22 | 551±82 | 52±14 | 106±15 | 192±167 | 515±118 | 83±15 | 554±161 |
| Benzoic acid, Ha | <DL | 5.4±2.1 | 1.9±0.2 | 3.8±0.3 | <DL | 42±57 | 2.5±0.4 | 4.0±1.1 | <DL | 12±7.8 | 3.1±0.3 | 6.0 ±2.0 |
| Total detected organics | 105±8.2 | 1329±695 | 127±54 | 1671±821 | 171±50 | 3530±626 | 241±76 | 1844±344 | 498±425 | 2626±583 | 439±90 | 4141±1166 |

[a] <DL denotes emissions below method detection limit (MDL).





Table 2 Comparison of mass ratios of $C_3/C_4$, $C_2/C_4$, $C_2$/total diacids, $\omega C_2/C_2$ and Gly/mGly in fresh and aged aerosols collected from biomass burning with the different locations around the World

| Category | Sampling site | Particle size | $C_3{}^1/C_4$ | $C_2/C_4$ | $C_2$/total diacids | $\omega C_2/C_2$ | Gly/mGly | References |
|---|---|---|---|---|---|---|---|---|
| Mountain | Mt. Hua | $PM_{10}$ | 2.0 | 10.7 | 0.6 | 0.06 | 0.6 | Meng et al. (2014) |
| | Mt. Tai | TSP | 0.8 | 5.3 | 0.6 | 0.1 | 0.5 | Kawamura et al. (2013) |
| | Mt. Fuji | TSP | 0.6 | 1.9 | 0.5 | 0.05 | 1.2 | Kunwar et al. (2019) |
| | Tokyo, Japan | TSP | 1.0 | 4.2 | 0.5 | 0.2 | 0.7 | Kawamura et al. (2005) |
| | Liaocheng, China | $PM_{2.5}$ | 0.4 | 3.6 | 0.6 | 0.1 | 1.0 | Meng et al. (2020) |
| Urban | Fairbanks | $PM_{2.5}$ | 1.2 | 4.2 | 0.5 | 0.1 | 1.4 | Deshmukh et al. (2018) |
| | DoiAngKhang, Thailand | $PM_{2.5}$ | 0.5 | 25.2 | 0.6 | 0.1 | 2.0 | Boreddy et al. (2021) |
| | Beijing, China | $PM_{2.5}$ | 0.8 | 6.8 | 0.5 | 0.1 | 0.6 | Zhao et al. (2018) |
| | Xi'an, China | $PM_{10}$ | 0.8 | 10.4 | 0.6 | 0.1 | 0.7 | Cheng et al. (2013) |
| | North Pacific | TSP | 1.4 | 5.3 | 0.5 | 0.01 | 2.0 | Kawamura et al. (1993) |
| Marine area | Eastern North Pacific | TSP | 1.1 | 4.3 | 0.5 | 0.004 | 0.2 | Hoque et al. (2020) |
| | Western North to equatorial Pacific | TSP | 3.9 | 14.3 | 0.6 | / | / | Kawamura et al. (1999) |
| Island | Okinawa | TSP | 1.9 | 15.5 | 0.8 | 0.06 | 0.5 | Kunwar et al. (2014) |
| | Motor Exhausts | | 0.35 | <1 | / | | | Kawamura et al. (1987) |
| | Siberian (biomass burning, chamber) | $PM_{2.5}$ | <0.03 | | / | / | / | Kalogridis et al. (2018) |
| Laboratory simulation | Fresh (biomass burning, chamber) | $PM_{2.5}$ | 0.2 | 0.7 | 0.1 | 1.3 | 3.8 | This study |
| | 2-day aged (biomass burning, chamber) | $PM_{2.5}$ | 0.3 | 3.8 | 0.6 | 0.3 | 2.3 | |
| | 7-day aged (biomass burning, chamber) | $PM_{2.5}$ | 0.5 | 6.4 | 0.6 | 0.2 | 2.0 | |


[1] See compund list in Table 1





Table 3 Stable carbon isotope ratios ($\delta^{13}$C, ‰) of $C_2$ in atmospheric aerosols from selected
locations

| Sampling site | Particle size | Min[1] | Max | Ave. | Std. | Sampling interval | References |
|---|---|---|---|---|---|---|---|
| **Urban** | | | | | | | |
| Liaocheng, China | PM$_{2.5}$ | -31.8 | -16.6 | -21.7 | 3.3 | Jan. to Feb. (Daytime) | Meng et al. (2020) |
| | PM$_{2.5}$ | -26.5 | -14.1 | -17.9 | 2.6 | Jan. to Feb. | |
| | PM$_{2.5}$ | -31.8 | -14.1 | -19.8 | 3.5 | Jan. to Feb. (Winter) | |
| Beijing, China | PM$_{2.5}$ | -23.7 | -15.0 | -20.1 | 3.0 | Sep. to Nov. | Zhao et al. (2018) |
| | PM$_{2.5}$ | -27.2 | -14.8 | -22.9 | 3.4 | Dec. to Feb. (Winter) | |
| | PM$_{2.5}$ | -25.0 | -16.6 | -21.9 | 2.1 | Mar. to May (Spring) | |
| | PM$_{2.5}$ | -27.0 | -19.1 | -22.4 | 2.7 | Jun. to Jul. (Summer) | |
| Sapporo, Japan | TSP | -22.4 | -14.0 | -18.8 | 2.0 | May to Jul. | Aggarwal et al. (2008) |
| **Marine** | | | | | | | |
| Western Pacific and Southern Ocean | TSP | -27.1 | -6.7 | -16.8 | 0.8 | Nov. to Feb. | Wang and Kawamura (2006) |
| **Mountain** | | | | | | | |
| Mt. Tai, China | PM$_{2.5}$ | -19.4 | -13.0 | -15.8 | 1.9 | Jul. to Aug. (Daytime) | Meng et al. (2018) |
| | PM$_{2.5}$ | -20.1 | -12.1 | -17.2 | 1.7 | Jul. to Aug. | |
| | PM$_{2.5}$ | -20.1 | -12.1 | -16.5 | 1.9 | Jul. to Aug. (Summer) | |
| **Background** | | | | | | | |
| Gosan, Korea | TSP | -15.0 | -10.6 | -12.6 | 1.4 | Mar. to May (Spring) | Zhang et al. (2016) |
| | TSP | -14.1 | -7.5 | -11.5 | 2.8 | Jun. to Aug. | |
| | TSP | -16.7 | -13.2 | -14.7 | 1.4 | Sep. to Nov. | |
| | TSP | -20.5 | -10.1 | -15.8 | 4.3 | Jan. to Feb. (Winter) | |
| **UV-irradiated** | | | | | | | |
| Ambient anthropogenic | PM$_{10}$ | | | -19.5 | | Non-irradiated | Pavuluri and Kawamura (2016) |
| | PM$_{10}$ | | | -13.1 | 3.6 | UV-irradiated | |
| Ambient biogenic aerosol | PM$_{10}$ | | | -15.8 | | Non-irradiated | |
| | PM$_{10}$ | | | -12.9 | 6.9 | UV-irradiated | |
| **This study** | | | | | | | |
| Maize straw | PM$_{2.5}$ | -14.9 | -12.1 | -13.1 | 1.6 | 2-day aged | This study |
| Rice straw | PM$_{2.5}$ | -28.2 | -24.6 | -26.2 | 1.8 | 2-day aged | |
| Wheat straw | PM$_{2.5}$ | -26.7 | -26.3 | -26.5 | 0.2 | 2-day aged | |
| Maize straw | PM$_{2.5}$ | -9.1 | -6.0 | -7.1 | 1.4 | 7-day aged | |
| Rice straw | PM$_{2.5}$ | -23.7 | -17.2 | -20.8 | 3.3 | 7-day aged | |
| Wheat straw | PM$_{2.5}$ | -24.6 | -23.5 | -24.0 | 0.5 | 7-day aged | |
| Biomass burning | PM$_{2.5}$ | -23.3 | -21.0 | -21.9 | 1.2 | 2-day aged | |
| | PM$_{2.5}$ | -19.1 | -15.5 | -17.3 | 1.7 | 7-day aged | |

[1]Min, Max, Ave, and Std stand for minimum, maximum, arithmetic mean, and standard deviation.