# Peer review of "Distribution and stable carbon isotopic composition of dicarboxylic acids, ketocarboxylic acids and α-dicarbonyls in fresh and aged biomass burning aerosols"

_Atmospheric Chemistry and Physics, 2021_

## Author Comment (AC1)

Journal: Atmos. Chem. Phys.

Title: Distribution and stable carbon isotopic composition of dicarboxylic acids, ketocarboxylic acids and  $\alpha$ -dicarbonyls in fresh and aged biomass burning aerosols

Author (s): Minxia Shen, Kin Fai Ho, Wenting Dai, Suixin Liu, Ting Zhang, Qiyuan Wang, Jingjing Meng, Judith C. Chow, John G. Watson, Junji Cao, Jianjun Li

MS No.: acp-2021-1049

Dear Editor,

After reading the comments from the two referees, we have carefully revised our manuscript. Our response and modifications to comments are listed in the attachment.

Anything about our paper, please feel free to contact me at lijj@ieecas.cn.

Best regards,

Sincerely yours

Jianjun Li

May 12, 2022

**Reviewer comments:**

**Reviewer #1:**

**General comment:**

This work studied the dicarboxylic acids (diacids) and related compounds from fresh and aged biomass burning (BB) source samples. The emission factors of each diacid and their carbon isotopic ratios were also calculated, which suggested the secondary formation nature of oxalic acid and related species. This work presents some interesting results given the uncertainty of direct emission of diacids from BB source samples. However, several issues need to be clarified before the consideration of acceptance.

**Response:** We thank the reviewer's comments, which are very helpful for us to improve our work. Detailed revision and response to the comments are list below.

1. The major concern is the setup of aging experiments of source samples in the chamber. Two and seven days are quite long for the oxidation of organics to diacids under high loadings of oxidant. Although authors had cited two references to support this setup, the detailed conversions from VOCs to oxalic acid were almost completed in two days from the ratios of 50.8 and 64.5 given in this work, so authors need to clarify this issue.

**Response:** It has been reported that 1- to 10-day aging period of most interest for determining PAM of peat burning plumes. In this study, the two  $OH_{exp}$  estimates were selected to examine intermediate (~ 2 day) and long-term (~ 7 day) atmospheric aging. As you pointed out, aged/fresh (A/F) ratio implies that two days of aging time may be sufficient for the formation of dicarboxylic acid (diacids). After 2-day aging, a large number of VOCs may have been oxidized and transferred to the particle phase by condensation, adsorption and other ways. For the longer aging times, the particulate phase compounds may be further oxidized to generate other compounds besides diacids. There were correlations between decreases in VOCs and increases in diacids from fresh to 2-day aged biomass burning samples, but no significant correlation was found 7-day aged, again confirming that two days of aging time was sufficient. Based on your comments, we have clarified this issue in the revised manuscript.

**Lines 44-49:**

However, no significant correlation was found between decreases in VOCs and increases in 7-day aged diacids. In addition, the A/F of  $C_2$  was 50.8 at 2 days and 64.5 at 7 days, indicating that the conversion of VOCs to  $C_2$  was almost completed within 2 days. For the longer aging times, the particulate phase compounds may undergo further degradation in the oxidation processes.

**Lines 265-277:**

In addition, we found that the A/F ratio of  $C_2$  after 2-day aging was 50.8, and the change from 2~ to 7 day was relatively small, only increasing by 13.7 (Table S1). These results meant that 2-day aging may be sufficient for most diacids formation. It can be

inferred that although diacids is still generated at 7-day aging, a large number of VOCs may have been oxidized at 2-day aging and transferred to the particle phase by condensation, adsorption and other ways. Especially for maize straw, the  $EF_{aged}$  of total detected organics at 7-day aging (1844 ± 344 mg/kg) was lower than that of at 2-day aging (3530 ± 626 mg/kg), which was mainly due to the predominant role of particulate diacids degradation in longer aging time. This phenomenon is consistent with the change of  $EF_S$  of VOCs (precursors of C2) during maize straw combustion. The decreases in of  $\Sigma$ VOCEF after 2-day aging (1227 mg/kg) were comparable with those of 7-day aging (884 mg/kg) for maize straw (Niu et al., 2020).

**Lines 417-419:**

Such a correlation between decreases VOCs and increases diacids again suggests that 2-day aging may be sufficient to oxidize VOCs to diacids.

**Lines 445-447:**

This correlation exists only at 2-day aging, but does not exist at 7-day aging, probably because the longer the aging time, the particle phase compounds may be further oxidized to other compounds.

2. Authors need to explain more clearly why the direct measurement of diacids from fresh BB samples is so important in the introduction in lines 83-86.

**Response:** Suggestion taken. The importance of direct measurement of diacid from fresh BB samples is described in more detail in lines 86-105 of the introduction in the revised manuscript.

**Lines 89-108:**

Diacids, ketocarboxylic acids and  $\alpha$ -dicarbonyls are products of BB (Agarwal et al., 2010). Although these acids have been measured in ambient air in some areas dominated by BB sources (FaLkovich et al., 2005; Kundu et al., 2010; Kawamura et al., 2013), there have been few BB sources emission (e.g., chamber) measurements. It was reported that BB smokes was found to contain large amounts of gaseous pollutants, such as VOCs, nitrogen oxides (NOx), sulfur dioxide (SO2), and ammonia (NH3) (Akagi et al., 2011; Andreae and Merlet, 2001). Gas-phase compounds, especially VOCs, can be partition to the particle phase through nucleation, condensation, and heterogeneous chemical reactions, creating secondary organic aerosol (SOA) and adding to aerosol mass (Hodshire et al., 2019; Lim et al., 2019). Oxalic acid (C2), the most abundant species of diacids (Kawamura and Sakaguchi, 1999) and is formed by various VOCs in cloud droplets through photochemical oxidation and liquid phase reactions. It is of interest to quantify emission factors (EFs) of diacids and related compounds during the combustion of different biomass fuels in the laboratory. Kalogridis et al. (2018) performed small-scale fire experiments using the Large Aerosol Chamber (LAC, 1800 m3) with a focus on BB from Siberian boreal coniferous forests, and presented experimental data on EFs of diacids. However, this study only focused

on the EFs of diacids of fresh pollutants that directly emitted from BB, so it is necessary to further investigation of molecular composition of aged BB aerosols.

3. Authors said "investigate reactions of volatile organic carbon compounds (VOCs) with oxalic acid and intermediates that form in the aging process" is one of the aims of this work. However, the detailed compositions and abundances of VOCs were not presented in the paper, and the supporting evidences were mainly from linear regressions between the reduction of VOCs and increase of oxalic acid. This was insufficient to give such important conclusion.

**Response:** We do agree with the reviewer's comment. Unfortunately, our current research cannot explain the reaction mechanism of VOCs and oxalic acid, and can only discuss the contribution of VOC to oxalic acid formation. Detailed compositions and abundances of fresh and aged VOCs (corresponding to 2- and 7-day atmospheric aging) from the burning of rice, maize, and wheat straws by a combustion chamber coupled with potential aerosol mass (PAM) reactor have been reported by Niu et al. (2020). The emission factors (EFs) of fresh and aged 56 kinds of VOCs (including acetylene, alkenes, alkanes, and aromatics) from different types of straw are given. Thus, the related sentences were rewritten as:

**Lines 120-121:**

**$\dots$ investigate relationship between VOCs with C2 and intermediates that form in the aging process to explore potential formation mechanisms of selected organic acids.**

Furthermore, in the section "Relationships between volatile organic carbon compounds and diacids", we have added a discussion of regression analysis (Lines 399-405) in addition to correlation. The results showed that the oxidation of VOCs led to the formation of diacids, which can be almost completed in 2-day aging.

**Lines 399-405.:**

Photooxidation of Gly and mGly is a major global and regional source of C2 diacid, and the two formation pathways are Gly- $\omega$ C2-C2 and mGly-Pyr- $\omega$ C2-C2, respectively (Yasmeen et al., 2010; Wang et al., 2012). As shown in the Fig.5, the slope (0.20~0.59) between the decrease of toluene and the increase of intermediates (Gly, mGly, Pyr and  $\omega$ C2) is significantly higher than C2 (0.04). Same thing with benzene, the slope between decrease of benzene and increase of mGly is 0.55, while C2 is only 0.05.

**4. How do you set the RH in the chamber study?**

**Response:** RH inside the OFR was varied by passing different amounts of carrier gas through the OFR humidifier (MH-110). The humidifier supplied with the reactor is configured to operate using the "siphon feed" method, in which sample/carrier gas is pushed through the inside of the Nafion membrane to the inlet of the reactor. RH was measured at the rear feedthrough plate with a RH and temperature probe (Sensirion SHT21, Switzerland). The specific location of the RH sensor is shown in Fig S2 attached to Cao et al. (2020). How to set RH has also been added to the revised manuscript.

Lines 154-155.:

Relative humidity (RH) inside the OFR was varied by passing different amounts of carrier gas through the OFR humidifier (MH-110).

**Minor comments:**

5. Line 230-231: rewrite.

Response: Suggestion taken.

Lines 260-261.:

These results provide further evidence that  $C_2$  is produced mainly through secondary photochemical processes rather than direct emission from BB.

6. Line 232: "This also is a likely reason", grammatical error.

Response: Suggestion taken.

Lines 261-262.:

"This also is a likely reason" was revised as "That is one possible reason".

7. Line 251-252: the" rapid" is contradictory to the description in lines 250-251.

**Response:** As for your doubts, I need to explain to you. As shown table, the EFs of  $C_4$  is higher than that of  $C_3$  in fresh, 2- and 7-day aging samples. However, the A/F ratio of  $C_3$  was much higher than that of  $C_4$  at 2 and 7-day aging. The reason may be attributed to rapid formation rate of  $C_3$  or decarboxylation processing of  $C_4$  diacid during aging (Zhao et al., 2018). There is something wrong with the previous expression. In the revised manuscript, the sentence is revised as follows:

Table R1 Emission factors (EFs) and ratios of aged to fresh (A/F) of malonic and succinic acid in fresh and aged PM2.5 aerosols

| Diacids                 |                            | Fresh     | 2-day aged | 7-day aged |
|-------------------------|----------------------------|-----------|------------|------------|
| Malonic, C 3 | EFs (mg kg -1 ) | 4.3 ± 3.7 | 69 ±18     | $133\pm40$ |
|                         | A/F                        | /         | 16.2       | 31.1       |
| Succinic, C4            | EF (mg kg -1 )  | 44 ± 25   | $252\pm65$ | 352 ± 109  |
|                         | A/F                        | /         | 5.7        | 8.0        |

**Lines 288-290.:**

[revised manuscript text omitted]

**Reviewer #2:**

This manuscript reported emission factors (EF) of dicarboxylic acids and related compounds from fresh and aged aerosols from burning rice, maize and wheat straw, and stable carbon isotope ( $\delta^{13}$ C) composition of these emissions. The authors provided a very interesting data, the EF of aged samples was significantly higher than that of fresh samples, and the molecular distribution was also significantly different. The carbon isotope of dicarboxylic acid was measured and the results showed that the  $\delta^{13}$ C value became more and more positive with the increase of aging degree. These results indicated that dicarboxylic acids and related compounds is largely produced by secondary photochemical processes rather than direct emissions in biomass burning. In my opinion, the MS can be eventually accepted for publication if below questions are adequately addressed.

**Response:** We also appreciate your comments and detailed revisions and responses to the comments are listed below.

1. In 2.1. "Preparation and collection of fresh and aged BB aerosols" section, more detail should be provided on the biomass burning. eg. Line 119~121, "A portion of the diluted smoke was drawn through a quartz fiber filter (47 mm diameter, Whatman QM/A, Maidstone, UK) at 5 L min-1 using a mini-Vol PM2.5 sampler (Airmetrics, OR, USA) to capture fresh emission…"PM2.5 sampler for collecting fresh samples is given in the manuscript. What is the sampler for PM2.5 aging aerosols? In which combustion phase was collected PM2.5 samples? How was the smoke diluted? Lines 121-124 "…and another portion (~9 L min-1) was drawn into a 19-L cylinder PAM-OFR (with a diameter of 20 cm and length of 60 cm) to simulate atmospheric aging." What is the residence time of aged biomass burning aerosol in OFR? Is the residence time of 2-day aging the same as that of 7-day aging?

**Response:** We thank the reviewer's suggestion. More details on biomass combustion are provided in "Preparation and Collection of fresh and aged BB aerosols".

Lines 123-166.:

**2.1. Preparation and collection of fresh and aged BB aerosols**

The experimental setup is illustrated in supplementary Fig. S1. Detailed procedures for sample preparation and collection may be found in previous studies (Li et al., 2020, 2021; Niu et al., 2020). Briefly, fresh smoke was generated by burning dry biomass fuels (i.e., rice, maize, and wheat straw) in a combustion chamber, and the smoke was then passed through a Potential Aerosol Mass-Oxidation Flow Reaction (PAM-OFR) (Aerodyne Research, LLC, Billerica, MA, USA) to simulate aging processes in timescale of hours to days. The biomass combustion chamber with a volume of ~8 m3 (1.8m (W)× 1.8m (L) ×2.2m (H)), which was made of 3 mm thick aluminum to withstand high-temperature heating. The combustion chamber was equipped with a thermoanemometer, an air purification system, a heated sampling line, a dilution sampler, and so on. More detailed information about the design and evaluation of combustion chamber were described in Tian et al. (2015).

In order to get sufficient aerosols samples for measurements of chemical composition, around 1 kg biomass fuels were burned inside the chamber in 10 burning cycles. The entire burning cycle, including ignition, flaming, smoldering, and extinction, intends to simulate real-world source characterization. Each burning cycle, containing  $\sim$ 100 g biomass fuels, lasts around 12 $\sim$ 18 min. The fresh smoke was diluted by 4.6 times using clean air controlled by the flow balance. A portion of the diluted smoke by dilution sampler (Model 18, Baldwin Environmental Inc., Reno, NV, USA) was drawn through a quartz fiber filter (47 mm diameter, Whatman QM/A, Maidstone, UK) at 5 L min-1 using a mini-Vol PM2.5 sampler (Airmetrics, OR, USA) to capture fresh emission.

The PAM-OFR can be used to simulate an environment with extremely high oxidant concentrations with short residence times (Kang et al., 2007). Another portion of the exhaust (~9 L min-1) was directed through a 19-L cylinder PAM-OFR (with a diameter of 20 cm and length of 60 cm) to simulate atmospheric aging. Residence time of PAM-OFR is estimated to be  $90 \pm 1$  s at flow rate of 9 L min-1 (Li et al., 2021). Three oxidants (O3, •OH, and •HO2) were generated in the PAM chamber using irradiation from ultraviolet (UV) lamps. The OH exposure values (OHexp) can be calculated by the concentration of SO2 and CO at the OFR inlet and outlet in a laboratory setting. Relative humidity (RH) inside the OFR was varied by passing different amounts of carrier gas through the OFR humidifier (MH-110). Additional details on smoke generation condition, test study and evaluation of the PAM-OFR were described by Cao et al. (2020).

In this study, the UV lamps operated at a voltage of 2 and 3.5 V,  $OH_{exp}$  in the chamber were estimated at 2.6 × 1011 and 8.8 × 1011 molecules-sec/m3, respectively. These levels corresponded to ~2 and 7 day of aging (Chow et al., 2019; Watson et al., 2019), assuming a representative atmospheric •OH level of  $1.5 \times 10^6$  molecules/m3 (Mao et al., 2009). The aged aerosols were sampled by another mini-Vol PM2.5 sampler (5 L min-1) following the reactions in the PAM-OFR chamber. Each test was conducted in triplicate to account for experimental errors and to provide a measure of variability, which was calculated as standard deviations. A total of 36 samples were collected and analyzed for chemical composition.

2. In 3.1. "Emission factors for dicarboxylic acids, ketocarboxylic acids,  $\alpha$ -dicarbonyls" section, Lines 201-205, C4 and C9 were most abundant in fresh biomass burning samples, why not C2? Please explain in detail.

**Response:** Actually, we have already explained this problem in the manuscript. First, previous studies showed  $C_9$  to be an oxidation product of unsaturated fatty acids in biomass smoke. Second, the results of aging experiment showed that  $C_2$  is produced mainly through secondary photochemical processes rather than direct emission from BB. Another possible reason has been added to the revised manuscript (Lines 240-242).

**Lines 240-242.:**

 $C_2$  is the most abundant species of diacids and is one of the final products of SOA reaction chain. In the fresh BB sample,  $C_2$  emissions were lower due to the short aging

**time.**

3. In 3.2. "Effects of atmospheric aging processes" section, EF*aged* of maize at 7 days was lower than that of at 2 days, which was different from that of wheat and rice straw. This phenomenon should be explained in this paper.

**Response:** We thank the reviewer's suggestion. An explanation of this phenomenon has been added to the revised manuscript.

**Lines 272-277.:**

Especially for maize straw, the EF*aged* of total detected organics at 7-day aging (1844  $\pm$  344 mg/kg) was lower than that of at 2-day aging (3530  $\pm$  626 mg/kg), which was mainly due to predominant role of particulate diacids degradation in longer aging time. This phenomenon is consistent with the change of EFs of VOCs (precursors of C2) during maize straw combustion. The decreases in of  $\Sigma$ VOCEF after 2-day aging (1227 mg/kg) were comparable with those of 7-day aging (884 mg/kg) for maize straw (Niu et al., 2020).

4. In 3.3. "Comparisons of diagnostic ratios of dicarboxylic acids in fresh and aged aerosols" section, Lines 300-301, Why is the degradation rate of Gly faster than that of mGly? How to reflect the consistency of Gly/mGly?

**Response:** The previous expression may have confused you, so we have rewritten this sentence in the in the revised manuscript.

**Lines 331-336:**

Aqueous-phase oxidation by OH is faster for Gly than for mGly, and the abundance of Gly relative to mGly is an indicator of aerosol aging (Cheng et al., 2013). The ratio of Gly/mGly in xi'an samples was lower in haze days than in clean days, and lower in summer than in winter. Similarly, the Gly/mGly ratios in the aged BB samples were higher in the fresh  $PM_{2.5}$  samples (3.8) compared to the 2-day (2.3) and 7-day (2.0) aging.

5. In 3.4. "Stable carbon isotopes" section, Lines 330-333, There was no significant difference in the isotopic data for simulating 2- and 7-day aging reactions. Do you think two days of aging simulation is enough?

**Response:** I think there is significant difference in the isotope for simulating 2- and 7day aging reactions. Except for wheat, the  $\delta^{13}$ C values of C2 in the 2-day and 7-day aging reactions are -26.5 ± 0.2 ‰ and -24.0 ± 0.5 ‰, respectively. The  $\delta^{13}$ C of C2 is more abundant in 7- than 2-day samples with -13.1 ± 1.6 ‰ (2-day) and -7.1 ± 1.4 ‰ (7-day) in maize; -26.2 ± 1.8 ‰ (2-day) and -20.8 ± 3.3 ‰ (7-day) in rice combustion. In addition, the A/F ratio in Fig. 3 showed that 2 days of aging is sufficient for many diacids. 6. In 3.5. "Relationships between volatile organic carbon compounds and dicarboxylic acids" section, the authors only discussed the correlation coefficient (R) between VOCs and dicarboxylic acids. The correlations are meaningful, but there is more information that can be interpreted from the regressions. Take a consideration at the regression analyses to see if they show anything interesting?

**Response:** We thank the reviewer's comment. Regression about the relationship between decreases in VOCs and increases in diacids were added in the revised manuscript (Lines 399-405).

**Lines 399-405.:**

Photooxidation of Gly and mGly is a major global and regional source of  $C_2$  diacid, and the two formation pathways are Gly- $\omega C_2$ - $C_2$  and mGly-Pyr- $\omega C_2$ - $C_2$ , respectively (Yasmeen et al., 2010; Wang et al., 2012). As shown in the Fig.5, the slope (0.20~0.59) between the decrease of toluene and the increase of intermediates (Gly, mGly, Pyr and  $\omega C_2$ ) is significantly higher than  $C_2$  (0.04). Same thing with benzene, the slope between decrease of benzene and increase of mGly is 0.55, while  $C_2$  is only 0.05.

7. In Conclusions, I think the conclusions ends a bit too abruptly, you should be to discuss what impact these results have on climate change or pollution control. In addition, in Line 386-388, The author mentioned "These results suggest that the dicarboxylic acids in the atmosphere largely originated from secondary photochemical processes as opposed to primary emissions from biomass burning." You should emphasize why this is a significant finding.

**Response:** We do agree with the reviewer's comments. The suggestion you mentioned has been added to the conclusion in the revised manuscript as follows:

**Lines 428-429.:**

It is confirmed for the first time whether the contribution of BB source to diacids is formed by primary emission or secondary oxidation.

**Lines 448-455.:**

Diacid are highly water-soluble in nature and thus their high abundances due to BB and intense photochemical aging would enhance the ability of aerosols to act as cloud condensation nuclei and modify the water-uptake properties of aerosol particles. Therefore, it is necessary to better understand the chemical and physical properties of the constituents of water-soluble organic smoke, as they may have a significant impact on climate forcings through indirect aerosol effects. The results provide in-depth understanding of secondary organic aerosol (SOAs) formation in regions greatly affected by BB.

8. During the 2-day and 7-day aging experiments, fresh samples were collected, and whether there were differences in the collection of fresh samples. As shown in Table 1,

EF results for fresh samples are different. Is the experiment reproducible?

**Response:** There was no difference in the collection of fresh samples during the 2-day and 7-day aging experiments. As shown in Table 1, although the  $EF_{fresh}$  of diacids, ketocarboxylic acids and  $\alpha$ -dicarbonyls are different, but they are on the same order of magnitude. In addition, the  $EF_{aged}$  of 2- and 7-day diacids approximately 10 times greater than the  $EF_{fresh}$ . Sufficient to illustrate the effect of aging process on the formation of diacids. Three replicates were set for each experiment and their standard deviations were calculated, indicating that the experiment was reproducible.

9. Some items of the manuscript are to be improved to avoid any errors in expression and stylistic phrases:

Line 28 "Biomass burning (BB) is a significant source for dicarboxylic acids (diacids) and..." The dicarboxylic acids are abbreviated as diacids, but the abbreviation is not used elsewhere in the abstract. Is it necessary to mention this abbreviation? Line 41 "...and the C2  $\delta^{13}$ C became isotopically heavier during aging." Line 230~231 "These results are further evidence that PM2.5 oxalic acid..."

**Response:** According to the reviewer's comment, we have revised the manuscript very carefully. For the specific changes, please see the revised manuscript. Thank the reviewer for this valuable suggestion.

**Lines 39-44.:**

[revised manuscript text omitted]

3812, https://doi.org/10.5194/acp-10-3803-2010, 2010.